# I KNOW THE FEELING: LEARNING TO CONVERSE WITH EMPATHY

## ABSTRACT

Beyond understanding what is being discussed, human communication requires an awareness of what someone is feeling. One challenge for dialogue agents is recognizing feelings in the conversation partner and replying accordingly, a key communicative skill that is trivial for humans. Research in this area is made difficult by the paucity of suitable publicly available datasets both for emotion and dialogues. This work proposes a new task for empathetic dialogue generation and EMPATHETICDIALOGUES, a dataset of 25k conversations grounded in emotional situations to facilitate training and evaluating dialogue systems. Our experiments indicate that dialogue models that use our dataset are perceived to be more empathetic by human evaluators, while improving on other metrics as well (e.g. perceived relevance of responses, BLEU scores), compared to models merely trained on large-scale Internet conversation data. We also present empirical comparisons of several ways to improve the performance of a given model by leveraging existing models or datasets without requiring lengthy re-training of the full model.

## 1 INTRODUCTION

Natural communication is frequently prompted by people sharing their feelings or circumstances. As examples, a recent study found that 80% of Twitter users seem to post mostly about themselves (Naaman et al., 2010), and ELIZA (Weizenbaum, 1966), one of the earliest chatbots developed, focused on asking its conversational partners why they were feeling a certain way. Interacting in these conversations requires reacting to what people share with an understanding of others' implied feelings. For instance, while the crossed-out response in Figure 1 is topically relevant, "Congrats! That's great!" is more natural because it acknowledges the underlying feelings of accomplishment.

Responding to people in a way that is empathetic or that acknowledges how the other person feels is a desirable trait for a dialogue agent, but still a non-trivial communicative skill. It is also currently difficult to measure. Although recent work has used large-scale corpora to train reasonably fluent and engaging dialogue agents (e.g. Mazaré et al. (2018)), existing chitchat dialogue benchmarks are not designed to capture whether those agents are responding in an empathetic way to implicit emotional cues. This may indeed be unlikely, given that the sources of Internet conversation data used for training are known to harbor aggressive and callous responses (Anderson, 2015).

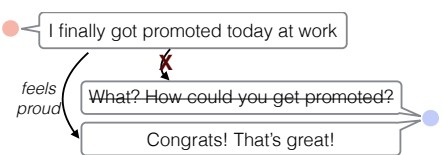

Figure 1: Example where acknowledging an inferred feeling might be appropriate

This works aims to make it easier to evaluate machines' ability to respond in an empathetic way. We introduce a new task for dialogue systems to respond to people discussing everyday situations, based on EMPATHETICDIALOGUES, a novel dataset with 25k personal dialogues. Each dialogue is grounded in a specific situation where a speaker was feeling a given emotion, with a listener responding. The dataset is larger and contains a more extensive set of emotions than many similar emotion prediction datasets from other text domains such as Scherer & Wallbott (1994), Strapparava & Mihalcea (2007), Mohammad et al. (2018), and Gupta et al. (2017). Previous dialogue datasets of a similar scale that include emotion labels (Li et al., 2017; Gupta et al., 2017) come from crawled

Table 1: Two examples from EMPATHETICDIALOGUES training set. The first worker (the speaker) is given an emotion label and writes their own description of a situation when they've felt that way. Then, the speaker tells their story in a conversation with a second worker (the listener).

| **Label: Afraid** | **Label: Proud** |
|---|---|
| **Situation:** Speaker felt this when... | **Situation:** Speaker felt this when... |
| "I've been hearing noises around the house at night" | "I finally got that promotion at work! I have tried so hard for so long to get it!" |
| **Conversation:** | **Conversation:** |
| Speaker: I've been hearing some strange noises around the house at night. | Speaker: I finally got promoted today at work! |
| Listener: oh no! That's scary! What do you think it is? | Listener: Congrats! That's great! |
| Speaker: I don't know, that's what's making me anxious. | Speaker: Thank you! I've been trying to get it for a while now! |
| Listener: I'm sorry to hear that. I wish I could help you figure it out | Listener: That is quite an accomplishment and you should be proud! |

conversations extracted from settings that are quite different from a one-on-one conversation (educational dialogues for English learners for DAILYDIALOG, public social media content for Gupta et al. (2017)) and cover either a very limited or a very imbalanced set of emotions: only $\approx 5\%$ of the DailyDialog utterances have a label other than 'none' or 'happy', and Gupta et al. (2017) only labels 'happy', 'sad', and 'angry'. The open resource we propose consists of crowdsourced one-on-one conversations, and covers a large set of emotions in a balanced way.

We then examine how to train a dialogue system that is more adept at responding to emotional cues. While a rule-based system can be built around mapping predicted emotions to responses, end-to-end dialogue systems relying on neural networks and trained on conversation corpora (Shang et al., 2015; Vinyals & Le, 2015; Sordoni et al., 2015; Serban et al., 2015; Dodge et al., 2016; Mazaré et al., 2018; Zhang et al., 2018) offer the promise of better generalization to new contexts. Through an extensive set of experiments, we show that fine-tuning a dialogue agent on our dataset results in better performance on a novel empathetic dialogue task.

The pretraining of the dialogue agent on Internet conversation data is the most time-consuming step of this pipeline, by an order of magnitude. To make it easier for practitioners to improve performance of a model on the empathetic task with minimal re-training while re-using existing resources, we compare various ways of supplementing a pretrained model with additional representations from external tasks, and show that even simplistic schemes can lead to better performance. The contributions of this work are thus threefold: 1) we release a novel empathetic dialogue dataset as a new benchmark; 2) we show that using this dataset for training can improve the performance of an end-to-end dialogue system on empathetic dialogue; and 3) we compare multiple ways to further improve performance with combined representations while not requiring onerous re-training.

## 2 RELATED WORK

Responding well to emotions requires sufficient coverage of human expression. Multiple schemas have attempted to organize the spectrum of emotions, from a handful of basic emotions derived from biological responses (Ekman, 1992; Plutchik, 1984) to larger sets of subtle emotions inferred from contextual situations (Skerry & Saxe, 2015). We incorporate emotions from multiple annotation schemas, noting that emotions merely inferred from a situation are important in dialogue scenarios.

Rich information can be represented by learning multidimensional distributional embeddings from data, as has proven successful for many language applications (Grave et al., 2018). These distributional representation approaches are at the core of the current state of the art in emotion classification (Duppada et al., 2018; Park et al., 2018; Xu et al., 2018; Mohammad et al., 2018) that build on deep networks pretrained on large-scale weakly labelled data such as emojis (Felbo et al., 2017) or hashtags (Mohammad, 2012), gathered from public social media content published on Twitter. The SEMEVAL2019 EmoContext challenge also uses conversation data for detection of three basic

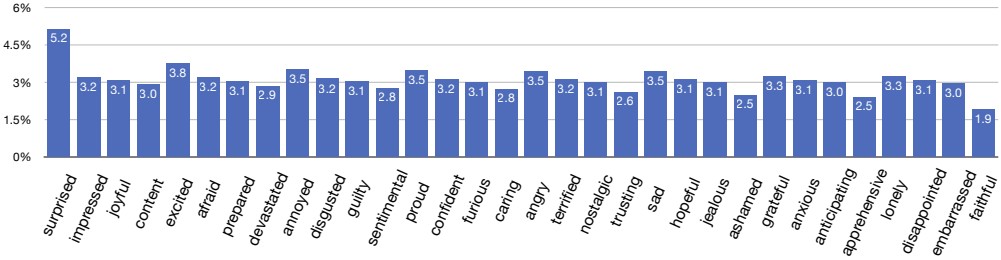

Figure 2: Distribution of situation/conversation labels within EMPATHETICDIALOGUES. Percentages per class are also listed in the appendix.

emotions over two turns of context from Twitter exchanges (Gupta et al., 2017). While public social media content has the advantage of being spontaneous (not elicited) data, it suffers from two shortcomings when used to train a model intended for one-on-one conversation (as opposed to, say, a bot designed to post on Twitter). First, the content is extracted from a context of communication in front of large "peripheral audiences" (Goffman, 1981) which include potentially everyone with an Internet connection, where the need for curated self-presentation (Goffman, 1959) and the uncertainty as to how wide that audience may be have been shown to lead to different choices of subject matters compared to private messaging, with people sharing more intense and negative emotions through private channels (Bazarova et al., 2015; Litt et al., 2014). Second, Tweets are generally a short-form format limited to 140 characters, which is not a constraint that applies to general conversation. In this work, we attempt to generate a more balanced coverage of emotions than would appear in public social media content, within a one-on-one framing of unconstrained utterances that is closer to our ultimate goal of training a model for conversation that can respond to any emotion.

Several works have attempted to make chit-chat dialogue models more engaging by grounding them in personal contexts (Li et al., 2016; Zhang et al., 2018; Mazaré et al., 2018), but focusing on personal facts ("I am from New York") rather than situations. The DAILYDIALOG dataset (Li et al., 2017), comprising about 13k dialogues obtained by crawling educational websites intended for learners of English, includes many dialogues anchored in everyday situations and has been annotated *post-hoc* with emotion labels, but only $\approx 5\%$ of the utterances have a label other than "none" or "happy", and dialogues are mostly limited to domains deemed appropriate for use as a language learning tool (ordering from a restaurant, asking for directions, shopping for a specific item, introductions, etc). Our task focuses explicitly on conversations about an emotionally grounded situation, and considers a richer, evenly distributed set of emotions. We also introduce an explicit single *listener* in the conversation who is reacting to the situation being described in an empathetic way, to make the setting as close as possible to our desired goal of a one-on-one empathetic conversation.

Several other works have focused on controlling the emotional content of a text response either through a manually specified target (Zhou & Wang, 2018; Zhou et al., 2017; Wang & Wan, 2018; Hu et al., 2017; Huang et al., 2018) or through a general term to encourage higher levels of affect (Asghar et al., 2018), with evaluations focused on matching a predetermined desired emotion rather than empathetic responding. Niu & Bansal (2018) generate responses conditioned on a specified politeness setting (polite, rude or neutral), where politeness is viewed as a style of language. Huber et al. (2018) investigate how to respond to emotions detected from an image. By contrast, our work examines how to produce empathetic responses that are appropriate to signals inferred purely from text, and not intended to themselves convey a pre-specified emotion.

## 3 TALKING ABOUT PERSONAL SITUATIONS

We consider an open-domain one-on-one conversational setting where two people are discussing a situation that happened to one of them and that led to the experience of a given feeling.

**Emotional situation grounding** Each conversation is grounded in a situation, which one participant writes about in association with a given emotion label. We consider 32 emotion labels, listed in Figure 2. To select this set of labels, we drew inspiration from previous datasets (Scherer & Wall-

bott, 1994; Strapparava & Mihalcea, 2007; Skerry & Saxe, 2015; Li et al., 2017; Mohammad, 2012), consolidating the labels from each into a merged list.

**Speaker and Listener**    The person who wrote the situation description (*Speaker*) initiates a conversation to talk about it. The other conversation participant (*Listener*) becomes aware of the underlying situation through what the Speaker says and responds. Speaker and Listener then exchange up to 6 more turns. We include two example conversations from the training data in Table 1. The models discussed below are tested in the role of *Listener* responding to the Speaker. The situation description generated by the Speaker is not given to the models (just as it was not given to the Listener during dialogue collection). Our data could also be used to generate conversations for the Speaker conditioned on the situation description; we leave this for later work.

**EMPATHETICDIALOGUES dataset statistics**    The resulting dataset comprises 24,850 conversations about a situation description, gathered from 810 different participants, which will be made publicly available online and through ParlAI. The crowdsourced dialogs were obtained using the ParlAI framework to interact with the Amazon Mechanical Turk (MTurk) platform. Details of the crowdsourcing procedure are given in the Supplemental Material.

The distribution of emotion label prompts (Table 2) is close to evenly distributed across categories with a few categories that are selected slightly more/less often. The average situation description is 19.8 words. Each conversation is allowed to be 4-8 utterances long (the average is 4.3 utterances per conversation). The average utterance length is 15.2 words long.

We split the conversations into approximately 80% train, 10% validation, and 10% test partitions. To prevent overlap of discussed situations by a same participant between partitions in case a participant chose to discuss the same situation twice, we split the data so that all sets of conversations with the same speaker providing the initial situation description would be in the same partition. The final train/val/test split was 19533 / 2770 / 2547 conversations, respectively.

## 4    EMPATHETIC DIALOGUE GENERATION

In this section, we examine how our dataset can be used to make generic chitchat models more empathetic, and different ways existing models can be combined to produce more empathetic responses. We use our dialogues to train and evaluate models in the task of generating conversation responses, with the model playing the *Listener* role. At test time, the dialogue model has access to previous utterances in the dialogue, but not to the emotion word prompt (e.g., "proud"), nor to the situation description generated by the Speaker, as would be the case in a normal conversation. Given a dialogue context $x$ of $n$ previous conversation utterances concatenated and tokenized as $x_1, \cdots, x_m$, followed by a target response $\bar{y}$, our models are trained to maximize the likelihood $p(\bar{y}|x)$ of producing the target response. We investigate both generation and retrieval settings (Lowe et al., 2016) as described in Figure 3.

### 4.1    BASE ARCHITECTURE

We base our models on Transformer networks (Vaswani et al., 2017), which have proven successful in machine translation and dialogue generation tasks (Zhang et al., 2018; Mazaré et al., 2018).

**Retrieval**    In the retrieval set-up, the model is given a large set $Y$ of candidate responses and picks the "best" one, $y^*$. We use the retrieval Transformer-based architecture from (Yang et al., 2018): two Transformer encoders separately embedding the context, $x$, and candidates, $y \in Y$, as $h_x$ and $h_y$, respectively. The model chooses the candidate sentence according to a softmax on the dot product: $h_x \cdot h_y$. We minimize the negative log-likelihood of selecting the correct candidate. At train time, we use all of the sentences from the batch as candidates, with a large batch size of 512 to give the model more negative examples. At inference time, we experiment with multiple sets of candidate sentences for the model to choose from. First, we use all of the response utterances in the EMPATHETICDIALOGUES training set ($Y^{ED}$). We also try including candidate utterances from two other large dialogue datasets: the DailyDialog (Li et al., 2017) training set ($Y^{DD}$) and up to a million utterances from a dump of 1.7 billion Reddit conversations ($Y^R$).

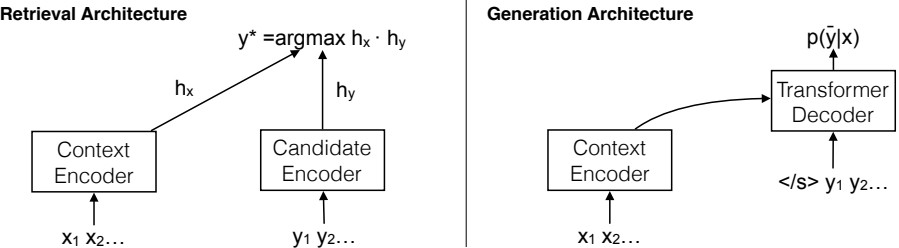

Figure 3: Dialogue generation architectures used in our experiments. The context of concatenated previous utterances is tokenized into $x_1, x_2, \cdots$, and encoded into vector $h_x$ by the context encoder. *Left:* In the retrieval set-up, each candidate $y$ is tokenized into $y_1, y_2, \cdots$ and encoded into vector $h_y$ by the candidate encoder. The system outputs the candidate $y^*$ that maximizes dot product $h_x \cdot h_y$. *Right:* In the generative set-up, the encoded context $h_x$ is used as input to the decoder to generate start symbol $$ and tokens $y_1, y_2, \cdots$. The model is trained to minimize the negative log-likelihood of target sequence $\bar{y}$ conditioned on context $x$.

**Generation**    In the generation set-up, we use the full Transformer architecture (Vaswani et al., 2017), consisting of an encoder and a decoder. The Transformer decoder uses the encoder output to predict a sequence of words $y$, and is trained to minimize the negative log-likelihood of the target sequence $\bar{y}$. At inference time, we use diverse beam search from Vijayakumar et al. (2016).

**Training Details**    We pretrain our models on predicting replies from a dump of 1.7 billion Reddit conversations. We limit the maximum number of word tokens in the context and response to be 100 each. The Transformer networks used in all experiments have the same base architecture (four layers and six transformer heads) from Mazaré et al. (2018), unless specified. For all models, we train for up to 10 epochs, keeping the version that has the lowest loss on the validation set. We use 300-d word embeddings pretrained on common-crawl data using fastText (Grave et al., 2018).

## 4.2 LEVERAGING THE TRAINING DATA FROM EMPATHETICDIALOGUES

**Conversations utterances**    Fine-tuning over the task domain data may improve the model, since our data was explicitly collected with instructions to be empathetic, in a one-on-one setting, which is different from the Reddit conversation data used for pretraining. We fine-tune pretrained models to predict the next utterance over our EMPATHETICDIALOGUES with a context window of four previous utterances, which is the average length of a conversation in our dataset. These models are hereafter referred to as "Base" models. This fine-tuning is conducted for all architectures except those referred to as "Pretrained".

**Emotion labels**    If the most appropriate response depends on some information for which supervision is available, e.g., the emotions at play, nudging the model to encode this information could result in better performance. We experiment with this by training the base architecture in the one-to-many style of multi-task learning that has been used for NLP seq2seq settings (Luong et al., 2016). In this set-up (Fig. 4, left), MULTITASK, we alter the objective function to also optimize for predicting the emotion label of the conversation to which the utterances being encoded belong. We add to the context encoder a linear layer and softmax that predicts the emotion label from the context sentences. The objective function is altered to be the average of the negative log-likelihood of predicting the next utterance $\bar{y}$ and the negative log-likelihood of the added linear layer being able to predict the correct emotion.

## 4.3 ADDING INFORMATION FROM EXTERNAL PREDICTORS

Many existing models have been pretrained on supervised tasks that may be relevant to empathetic responding. For instance, Felbo et al. (2017) have released a model trained on more than a billion tweets to predict emoji labels. Combining these models with the representation of our base architecture may reap benefits from previous training time and external training data without having to

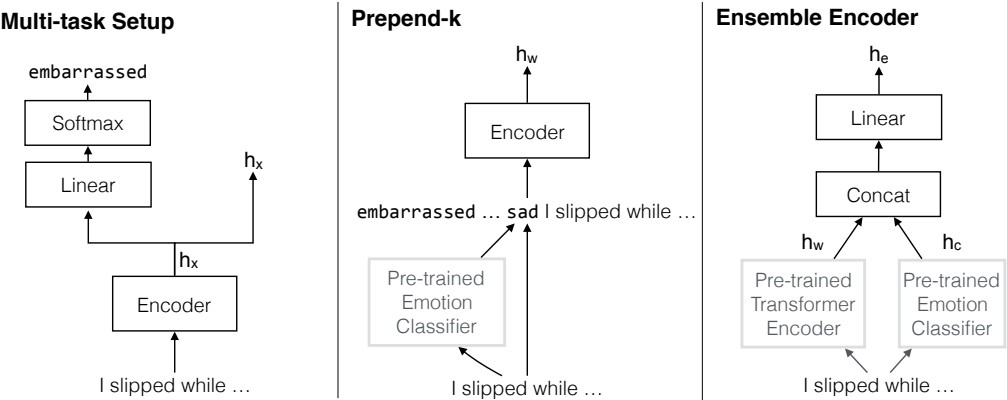

Figure 4: Three ways to incorporate additional supervised information, here from an emotion classification task. *Left*: the context representation $h_x$ outputted by the context encoder is used both as input to a classifier, and to generate the next utterance as in the base setting. The encoder is trained with gradients from both output branches. *Middle:* an input sequence (that can be either a dialogue context or a candidate) is first run through a pretrained classifier, and the top $k$ output labels are prepended to the sequence, which is then run through the corresponding (context or candidate) encoder to output a hidden representation $h_w$ (either $h_x$ or $h_y$) as in the base setting. *Right:* an input sequence (that can be either a dialogue context or a candidate) is run through the corresponding encoder as well as a pretrained classifier with the last layer removed. The outputs $h_w$ and $h_c$ are concatenated and linearly projected into a representation $h_e$.

redo the work or requiring access to that data, which may matter to practitioners. Note that this may considerably augment the effective capacity of the resulting models, as well as the total amount of training data used overall, so it isn't surprising that this should result in better performance. Our goal here is to experiment with a large range of settings, in order to get an empirical sense of how robust performance improvement is to variations in architecture set-up or supervision domain. We experiment with two set-ups for adding explicit supervised information: prepending predicted label words, and ensemble learning over encoders trained on prediction objectives.

**Prepending Top-K Predicted Labels** This set-up (Fig. 4, middle), PREPEND-K, is a very simple way to add supervised information to data, requires no architecture modification, and can be used with black-box classifiers. The top-K predicted labels from the supervised classifier are merely prepended to the beginning of the token sequence as encoder input:

I finally got promoted! $\longrightarrow$ `proud excited joyful` I finally got promoted!

Similar methods have been used for controlling the style of generated text (e.g. Niu & Bansal (2018)). Here, we use a fastText model (Joulin et al., 2017) as prediction architecture. Both the context and the candidates are run through the classifier and receive prepended labels. We experiment with two sources of supervision (EMOPREPEND and TOPICPREPEND, respectively). Closest to our task, we train a classifier to predict the emotion label from the description of the situation written by the Speaker before the dialogue for the training set dialogues of our EMPATHETICDIALOGUES.[1] To gauge whether supervision from a more distant task would still be helpful, we also experiment with a classifier trained on the 20-Newsgroup dataset (Joachims, 1996), for topic classification.

**Ensemble of Encoders** In this set-up (ENSEM), we augment the encoders to incorporate latent representations from pretrained supervised architectures. We replace each of the encoders in our Transformer networks with the Ensemble encoder in Figure 4, similar to a many-to-one style encoder-decoder arhcitecture (Luong et al., 2016). This encoder takes the encoding $h_w$ from our basic Transformer encoder (either $h_x$ or $h_y$), already trained on our data, and concatenates it with the representation $h_c$ extracted from the inner layer of a classification network. Here, we use the penultimate layer of a deep classifier. The concatenated encodings are projected linearly to the dimension

---

[1] We also experimented with training the classifier on the utterances themselves, with similar results.

Table 2: Automatic evaluation metrics on the test set. Pretrained: basic Transformer model pre-trained on a dump of 1.7 billion REDDIT conversations. Base and Multitask: model fine-tuned over the EMPATHETICDIALOGUES training data (Sec. 4.2). Remaining rows: models incorporating supervised information from an external classifier, as described in Sec. 4.3. Candidates come from REDDIT (R), EMPATHETICDIALOGUES (ED), or DAILYDIALOGUES (DD). P@1,100: precision retrieving the correct test candidate out of 100 test candidates. AVG BLEU: average of BLEU-1,-2,-3,-4. PPL: perplexity. All automatic metrics clearly improve with in-domain training on utterances (Base vs. Pretrained), other metrics are inconsistent. Base model row is repeated as it is the reference for the external classifier group. *Bold: best performance for that group.*

| Model | Retrieval | | | Generation | |
| | P@1,100 | Candidate Source | AVG BLEU | PPL | AVG BLEU |
| --- | --- | --- | --- | --- | --- |
| Pretrained | 43.25 | R | 4.1 | 27.96 | 5.01 |
| | - | ED | 5.51 | - | - |
| Base | **56.90** | ED | 5.88 | **21.24** | **6.27** |
| | - | ED+DD | 5.61 | - | - |
| | - | ED+DD+R | 4.74 | - | - |
| Multitask | 55.73 | ED | **6.18** | 24.07 | 5.42 |
| Base | **56.90** | ED | 5.88 | 21.24 | 6.27 |
| EmoPrepend-1 | 56.31 | ED | 5.93 | 24.30 | 4.36 |
| EmoPrepend-3 | 55.75 | ED | **6.23** | 23.96 | 2.69 |
| EmoPrepend-5 | 56.35 | ED | 6.18 | 25.40 | 5.56 |
| TopicPrepend-1 | 56.38 | ED | 6.00 | 25.40 | 4.17 |
| TopicPrepend-3 | 55.44 | ED | 5.97 | 25.02 | 3.13 |
| TopicPrepend-5 | 55.75 | ED | 6.17 | 25.10 | 6.20 |
| Ensem-DM | 52.71 | ED | 6.03 | **19.05** | **6.83** |
| Ensem-DM+ | 52.35 | ED | 6.04 | 19.1 | 6.77 |
| Ensem-Tran | 51.69 | ED | 5.88 | 19.21 | 6.41 |

required by the decoder, whose architecture doesn't change. When training the dialogue model, we freeze both the base Transformer encoder and the pretrained classifier (grayed out in Figure 4), and train only the linear layers (and the decoder for generative systems). We used supervision from two sources that are related to emotion: Emojis from Twitter, through the use of the trained Deep-moji system (Felbo et al., 2017) released by the authors, either as-is (ENSEM-DM) or fine-tuned on the situation descriptions of EMPATHETICDIALOGUES(ENSEM-DM+), and a similarly large-scale dataset of public social media content labelled by their writers with emotion tags such as 'annoyed', used to train a second Transformer encoder ( ENSEM-TRAN).

## 5 EXPERIMENTAL EVALUATION

We evaluate the models on their ability to reproduce the Listener's portion of the conversation (i.e. the ability to react to someone else's story). We use both automated metrics and human evaluation to score each model's retrievals/generations. Human evaluation is useful, as automated metrics don't always correlate with human judgments of dialogue quality (Liu et al., 2016), but we provide automated metrics to give a sense of how well they align with human judgment on this task.

**Automated Metrics (Table 2)** For both retrieval and generative systems, we compute BLEU scores (Papineni et al., 2002) for the final response and compare against the gold label (the actual response), following the practice of earlier work in dialogue generation (Wen et al., 2015; Li et al., 2015; 2016). For the generative systems, we additionally report perplexity of the actual gold response. For the retrieval systems, we additionally compute `p@1,100`, the accuracy of the model at choosing the correct response out of a hundred randomly selected examples in the test set. When

Table 3: Human evaluation metrics from rating task. Training on EMPATHETICDIALOGUES improves all scores. Combining with an external supervised classifier generally improves scores, especially the Empathy score, without requiring extensive retraining of the dialogue model. The Base models rows are repeated to indicate that they are the reference model for the models below. *Bold: results above 2 SEM of reference model for that group. Italicized: the reference model for the group*

|  | Model | Candidates | Empathy | Relevance | Fluency |
|---|---|---|---|---|---|
| Retrieval | *Pretrained* | R | *2.58±0.14* | *2.97±0.14* | *4.11±0.12* |
|  | Base | ED | **3.27±0.13** | **3.42±0.14** | **4.44±0.08** |
|  | Multitask | ED | **3.58±0.12** | **3.58±0.14** | **4.46±0.09** |
|  | *Base* | ED | *3.27±0.13* | *3.42±0.14* | *4.44±0.08* |
|  | EmoPrepend-1 | ED | 3.51±0.13 | 3.61±0.15 | 4.45±0.10 |
|  | EmoPrepend-3 | ED | **3.62±0.14** | 3.50±0.15 | 4.54±0.08 |
|  | EmoPrepend-5 | ED | 3.52±0.14 | 3.64±0.14 | 4.47±0.09 |
|  | TopicPrepend-1 | ED | **3.66±0.11** | **3.85±0.11** | 4.51±0.08 |
|  | TopicPrepend-3 | ED | **3.67±0.10** | **3.70±0.11** | 4.49±0.08 |
|  | TopicPrepend-5 | ED | **3.59±0.10** | **3.73±0.10** | 4.43±0.08 |
|  | Ensem-DM+ | ED | 3.36±0.14 | 3.33±0.14 | 4.13±0.11 |
|  | Ensem-Tran | ED | **3.80±0.12** | 3.66±0.14 | 4.59±0.08 |
| Generation | *Pretrained* | - | *2.26±0.13* | *2.37±0.13* | *4.08±0.12* |
|  | Base | - | **2.95±0.15** | **3.10±0.14** | **4.37±0.10** |
|  | Multitask | - | **3.17±0.14** | **3.23±0.14** | 4.29±0.11 |
|  | *Base* | - | *2.95±0.15* | *3.10±0.14* | *4.37±0.10* |
|  | EmoPrepend-1 | - | 2.66±0.15 | 2.63±0.15 | 4.22±0.12 |
|  | EmoPrepend-3 | - | **3.34±0.13** | 3.31±0.15 | **4.58±0.09** |
|  | EmoPrepend-5 | - | **3.35±0.15** | 3.20±0.15 | 4.41±0.10 |
|  | TopicPrepend-1 | - | 3.05±0.11 | 3.19±0.12 | 4.31±0.09 |
|  | TopicPrepend-3 | - | 3.12±0.11 | 3.10±0.13 | 4.33±0.09 |
|  | TopicPrepend-5 | - | **3.45±0.12** | **3.39±0.11** | 4.49±0.08 |
|  | Ensem-DM+ | - | 3.17±0.14 | 3.19±0.14 | 4.31±0.11 |
|  | Ensem-Tran | - | **3.49±0.12** | 3.27±0.14 | 4.42±0.09 |
| *Gold Response* | - | - | *4.19±0.06* | *4.48±0.06* | *4.67±0.04* |

we compute `p@1,100`, the actual response is included in the candidates, unlike inference from the retrieval systems for all other metrics, which only uses training utterances as candidates.

**Human Ratings (Table 3)** We run two sets of crowdsourcing tasks on MTurk for humans to score the model responses. In the first task, participants are given a model's output for a randomly selected test set example and asked to score different aspects of the model. The rating task provides a means of comparing aspects of responses, and we are able to ask raters specifically about whether the response is acknowledging the conversation partner's feelings. We collected at least 100 ratings per model and asked about three aspects of performance, all rated on a likert scale (1: not at all, 3: somewhat, 5: very much):

- Empathy/Sympathy: did the responses show understanding of the feelings of the person talking about their experience?

- Relevance: did the responses seem appropriate to the conversation? Were they on-topic?

- Fluency: could you understand the responses? Did the language seem accurate?

**Human A/B Rankings (Table 5)** In the second human evaluation task, participants were given output from two (randomly ordered) models and asked to select the better response, with an additional option of selecting "equal" or "neither". For this task, we only gave workers test set examples where the pair of models had differing responses. We collected at least 50 ratings per pair of models.

Table 4: Training resources for different models, with human evaluation metrics from rating task for empathy (Emp), relevance (Rel) and fluency (Fluent). Comparisons are relative to the first row of each group. Training on EMPATHETICDIALOGUES improves all scores while requiring minimal additional training resources (same number of parameters for Base, 0.01% increase in number of parameters for Multitask). SEM is approximately 0.1

|  | Model | Params, resources, train examples | Emp | Rel | Fluent |
|---|---|---|---|---|---|
| Retrieval | Pretrained | 84.3M, 2.5 days, 8GPUs, 1.7B | 2.6 | 3.0 | 4.1 |
| | Base | same , + 0.5 hour, 1 GPU, +22.3k | 3.3 | 3.4 | 4.4 |
| | Multitask | +9.6k, + 0.5 hour, 1 GPU, +22.3k | 3.6 | 3.6 | 4.5 |
| Generation | Pretrained | 85.1M, 2 days, 32 GPUs, 1.7B | 2.3 | 2.4 | 4.1 |
| | Base | same , +1 hour, 1 GPU, +22.3k | 3.0 | 3.1 | 4.4 |
| | Multitask | +9.6k, +1 hour, 1 GPU, +22.3k | 3.2 | 3.2 | 4.3 |
| | PretrainedLarge | 86.2M, 2.5 days, 32 GPUs, 1.7B | 3.0 | 3.1 | 4.0 |
| | BaseLarge | same , +0.5 hour, 1 GPU, +22.3k | 4.0 | 4.2 | 4.7 |

## 5.1 RESULTS

**Fine-tuning on EMPATHETICDIALOGUES**  Table 2 shows that fine-tuning to predict conversation responses on our data improves all automated metrics. Using only in-domain candidates leads to slightly higher BLEU scores. Training in the multitask setting degrades automated metrics compared to fine-tuning without emotion label supervision, except for average BLEU in the retrieval setting. Human evaluations in Table 3 show that fine-tuning a conversational model on the EMPATHETICDIALOGUES data and using the candidates in the dataset substantially improves performance on all metrics, in particular on the Empathy subscore of most interest to us, in both retrieval and generation set-ups. The bulk of the improvement comes from fine-tuning on the dataset, while results from the multitask setting suggest slight improvements in the Empathy rating compared to the model fine-tuned with utterances only.

**Resources and capacity**  Fine-tuning the pretrained conversation model on dialogue utterances from our data does not change the size of the model, so the performance improvement is not due to increased model capacity. However, the multitask setting slightly increases the capacity of the base architecture (about 10k more parameters out of 84.3M or 85.1M for the retrieval and generative architectures, respectively), which may account for the performance improvement. To give a better sense of how resources and capacities balance out, Table 4 provides figures of resource use and number of parameters. We also include results for a larger Transformer model (5 layers instead of 4). A generative model trained on our data performs as well or better than a generative pretrained Transformer model with one extra layer, which has a substantially larger capacity (a million more parameters), but not as well as such a model after it has been fine-tuned on our data.

**Augmenting conversation models with external pretrained classifiers**  Table 2 shows automated metrics for retrieval models augmented with external classifiers. Generally, the accuracy of the rankings (p@1,100) worsens, but the average BLEU scores improves. For generative models, the automated metrics are improved only in the Ensemble setting. Human evaluations (Table 3) suggest that nearly all models tend to improve over the Base model on the Empathy score, but only a few set-ups lead to significant improvement. The best performance is achieved by the ensemble trained on emotion tags (Ensem-Tran), but models that prepend topics are overall performing better than similar models trained on emotion supervision, or than the fine-tuned DeepMoji ensemble. Relevance is significantly improved only by models augmented with topic classifiers. Fluency scores are mostly not significantly changed, and remain above 4. The benefit of combining pretrained models is that this requires minimal re-training of parameters and allows for leveraging models that are already trained, or that can be trained in a few hours in the case of fastText classifiers. If re-training is not a concern, pre-training a larger Transformer model and fine-tuning it on our data leads to better performance (see Table 4). We observe that augmenting that model with pre-trained unsupervised sentence representations in an ensemble setting could also be a promising direction, but we leave that for future investigations.

Table 5: Average ratio of "best" replies from model A vs. model B for a set of pairs in our human ranking evaluation. Ratios > 1 mean that the model on the left was selected more than the model on the right. Emotion Supervision Models: models from Sec. 4.3 that incorporate supervised emotion information. Full listings of comparison between pairwise models is included in the appendix.

| Choose Model A vs B response | Average A:B ratio |
|---|---|
| Gold Response vs. Models | 13.79 |
| Generation vs. Retrieval | 0.62 |
| Emotion Context Models vs Pretrain (ret) | 3.67 |
| Emotion Context Models vs Pretrain (gen) | 5.73 |
| Emotion Context Models vs Base (ret) | 0.97 |
| Emotion Context Models vs Base (gen) | 1.56 |

Table 6: Examples of model responses. In the first example, responses from the models with emotion prediction components focusing on the feelings of the speaker are more generic. In the second example, they focus on the feelings on the speaker while remaining topically specific.

| Context | Model | Generated Utterance |
|---|---|---|
| Tomorrow is my first university day and I'm very nervous | Gen-base | What are you studying? |
| | Gen-multitask | Good luck! |
| | Gen-emoPrepend-5 | I'm sure you will be fine. |
| | Gen-ensemb-DM+ | Are you nervous ? |
| I had to go out of town by myself for three weeks. It was hard not being able to talk to anyone I knew. | Ret-base | Do you still talk to the people you went with? |
| | Ret-multitask | I'm sorry to hear that. Do you think you struggle to communicate with people? |
| | Ret-emoPrepend-5 | That sounds difficult. Did you try talking to people in your neighborhood? |
| | Ret-ensemb-DM+ | Did you start to feel a bit lonely? |

**A/B comparisons** To try and capture the main takeaways, we show results averaged over pairs of models with similar characteristics (Table 5). Responses from retrieval systems are frequently chosen over generation systems (generation:retrieval ratio: 0.62). Responses from models with added emotion supervision are often ranked above the raw pretrained model (ratios of 3.67 and 5.73), and less so against the base Transformer model fine-tuned on our data: in the retrieval case, the ratio of picking a line from a model with emotion supervision vs. base model, at 0.97, indicates that raters generally picked from each equally. However, in the generation case, the raters may have favored the models with emotion supervision explicitly represented (average ratio of 1.56).

## 6 CONCLUSION

We introduce a new dataset of 25k dialogues grounded in situations prompted by specific emotion labels. Our experiments show that using this dataset to fine-tune conversation models leads to responses that are evaluated by humans as more empathetic, and that simple schemes to augment a fine-tuned model with an external pretrained classifier can lead to better performance without requiring onerous retraining. Future work will investigate how to use this dataset to model the Speaker, and how to integrate empathetic responding into more general dialogue when, for example, the needs for empathy have to be balanced with staying on topic or providing information (see Table 6). Other possible directions would be to see if this data can serve as additional weakly supervised data for more complex emotion-related tasks that look at emotion evolution or causality (Gui et al., 2016; Rashkin et al., 2018). We hope that our results and dataset will stimulate more research in the important direction of making dialog systems more empathetic.

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

Table 7: Distribution of situation/conversation labels within EMPATHETICDIALOGUES

| Label | % | Label | % | Label | % | Label | % |
|---|---|---|---|---|---|---|---|
| surprised | 5.15 | impressed | 3.24 | joyful | 3.10 | content | 2.97 |
| excited | 3.77 | afraid | 3.21 | prepared | 3.08 | devastated | 2.87 |
| annoyed | 3.53 | disgusted | 3.17 | guilty | 3.06 | sentimental | 2.81 |
| proud | 3.49 | confident | 3.16 | furious | 3.05 | caring | 2.75 |
| angry | 3.47 | terrified | 3.15 | nostalgic | 3.05 | trusting | 2.62 |
| sad | 3.45 | hopeful | 3.14 | jealous | 3.05 | ashamed | 2.53 |
| grateful | 3.28 | anxious | 3.11 | anticipating | 3.03 | apprehensive | 2.45 |
| lonely | 3.28 | disappointed | 3.10 | embarrassed | 2.98 | faithful | 1.92 |

## A  SUPPLEMENTAL MATERIAL

### A.1  LABEL DISTRIBUTION

In Table 7, we include the exact percentage of emotion labels for the situation descriptions in our final dataset.

### A.2  DIALOGUE COLLECTION: CROWDSOURCING DESCRIPTION

We collected crowdsourced dialogues using the ParlAI platform (Miller et al., 2017) to interact with Amazon Mechanical Turk (MTurk). A pair of workers are asked to (i) select an emotion word each and describe a situation when they felt that way, and (ii) have a conversation about each of the situations, as outlined below.

**Writing a situation description prompted by an emotion label**  In the first stage of the task, workers are asked to describe in a few sentences a situation based on a feeling label. Each worker is given three labels from our list of 32 emotions. They are asked to select one of the options and write a short description of a personal situation where they felt that way. We ask the workers to try to keep these descriptions between 1-3 sentences. The average response is 19.8 words.

**Having a conversation**  In the second stage, two workers are paired and asked to have two short chats with each other. In each chat, one worker (*speaker*) starts a conversation about the situation they previously described, and the other worker (*listener*) responds. Neither can see what the other worker was given as emotion label or the situation description they submitted, so they must respond to each others' stories based solely on cues within the conversation. Each conversation is allowed to be 4-8 utterances long (the average is 4.31 utterances per conversation). The average utterance length was 15.2 words long.

**Ensuring balanced prompt coverage**  After the first few initial rounds of data collection, we forced workers to select prompts among three emotion labels that had been the least chosen overall so far if it was their first time working on the task. If they had already performed the task, the offered emotion labels were among those that they had not chosen before, or among the three least chosen ones if they had worked on nearly all of them at least once. This process made workers select emotions that they might not spontaneously have preferred, but we observed an initial bias for situations that were easier to describe (e.g, a situation causing surprise). Given that a conversation model trained for empathetic responding needs to be able to handle emotions even if they are less frequent, we opted for this balancing procedure to make training for these categories easier, while still allowing for some measure of choice for workers.

**Workers**  810 US workers were recruited using MTurk. Each worker had to contribute at least one situation description, and one pair of conversations: one as Speaker about the situation they contributed, and one as Listener about the situation contributed by another worker. Workers were allowed to accept additional hits and contribute more sets of situation descriptions, conversations as Speaker, conversations as Listener. The median number of conversation per worker was 8, while

Table 8: 10 random examples from EMPATHETICDIALOGUES training set.

**Label: Content**
**Situation:** Speaker felt this when...
"eating my favorite meal makes me happy."
**Conversation:**
Speaker: i am at my best when i have my favorite meal.
Listener: nice
Speaker: i love enchiladas
Listener: really?
Speaker: yes. enchiladas for the win!

**Label: Proud**
**Situation:** Speaker felt this when...
"I was proud when my brother finished college. He worked so hard at it"
**Conversation:**
Speaker: I was proud of my brother when he finished school. He worked so hard at it
Listener: Nice, tell him congrats. What did he major in?
Speaker: It was English
Listener: He should become an English teacher1

**Label: Joyful**
**Situation:** Speaker felt this when...
"I have had a great week!"
**Conversation:**
Speaker: I have had a great start to my week!
Listener: That's great. Do you think the rest of the week will be as great?
Speaker: I hope so! It looks promising!!
Listener: Lucky you. Are you always a positive person or it's just been an amazing week really?
Speaker: haha. Kind of both. And also probably too much coffee to start my shift tonight

**Label: Terrified**
**Situation:** Speaker felt this when...
"I got home for lunch and found a bat outside on my front porch."
**Conversation:**
Speaker: I got home for lunch and found a bat outside on my front porch. It probably has rabies. Bats shouldn't be out during the day.
Listener: Doesn't rabies cause sensativity to light? Either way I would freak out...
Speaker: It can but, it also causes anmails to behave erratically... like bats wadering around in the middle of the day.
Listener: Oh yeah, gotcha. I really don't like animals that are small and move quickly
Speaker: Generally yes.

**Label: Anticipating**
**Situation:** Speaker felt this when...
"I cant wait to go on my end of summer trip"
**Conversation:**
Speaker: I cant wait to go on my end of summer trip in texas.
Listener: Sounds like fun. What you got planned ?
Speaker: not really sure but im excited to just be invited
Listener: Got any family out there? Cousins perhaps

**Label: Terrified**
**Situation:** Speaker felt this when...
"My brother jump scared me while I was out playing. It was crazy bad."
**Conversation:**
Speaker: Just got scared to death.
Listener: Oh no. What happened?
Speaker: My brother jumped scared me.
Listener: lol is he younger or older?

**Label: Proud**
**Situation:** Speaker felt this when...
"My little dog learned to sit!"
**Conversation:**
Speaker: I finally tough my new little puppy his first trick!
Listener: What trick did you teach him?
Speaker: I tought him to sit for a treat, its so cute.
Listener: That is good, do you plan to teach him more tricks?

**Label: Apprehensive**
**Situation:** Speaker felt this when...
"I have to call my landlord about being late on the rent. I really don't want to have this conversation."
**Conversation:**
Speaker: I have to make a dreadful phone call tomorrow
Listener: Oh no, about what?
Speaker: I'm late on my rent and I need another week. I don't want to because my landlord isnt very nice
Listener: Oh no, I've been there done that too many times.
Speaker: I don't want her to make a big deal

**Label: Confident**
**Situation:** Speaker felt this when...
"When my husband asked me about how to build a chicken coop I was able to give him a reply that was backed up by blueprints and research from the internet. "
**Conversation:**
Speaker: We recently got 9 chicks and we've been having to work on making them a coop! I had to do so much research but I think we finally have a place that they'll enjoy living when they aren't able to free range.
Listener: OHH! I Love chickens ! I have always wanted some. I have a duck! lol- What kind of chickens are they?
Speaker: We currently have 2 Australorps, 3 Rhode Island Reds, 3 Barred Plymouth Rocks, and 1 Welsummer, but 4 of the 9 ended up being roosters. Ugh!
Listener: Oh man! They fight sometimes. I hope they aren't too bad about waking you up in the morning. Chickens can be very sweet though!
Speaker: I love my little hens, especially one I've named Curly. The roosters might get replaced by hens though because the crowing is so frustrating!

**Label: Surprised**
**Situation:** Speaker felt this when...
"I got a lottery ticket while I was at work today. I won $100 on the scratch off. I was shocked. I never win."
**Conversation:**
Speaker: I won $100 on a scratch off today. I was shocked. I never win.
Listener: Wow! How often do you play the lottery?
Speaker: I usually go on our Tuesday break to buy one with coworkers.
Listener: Neat! Well that is a fantastic feat. Maybe you can win again sometime?

Table 9: Classification performance on EMPATHETICDIALOGUES, with the benchmarks proposed in Felbo et al. (2017) for reference. ED: performance on predicting the emotion label from the situation description. ED-CUT: same, but after having removed all the situation descriptions where the target label was present.

| Dataset | Metric | SOTA (in 2017) | fastText | DeepMoji new | DeepMoji full | DeepMoji last | DeepMoji chain-thaw |
|---|---|---|---|---|---|---|---|
| SE0714 | F1 | 0.34 | 0.16 | 0.21 | 0.31 | 0.36 | 0.37 |
| OLYMPIC | F1 | 0.50 | 0.38 | 0.43 | 0.50 | 0.61 | 0.61 |
| PSYCHEXP | F1 | 0.45 | 0.44 | 0.32 | 0.42 | 0.56 | 0.57 |
| SS-TWITTER | Acc | 0.82 | 0.68 | 0.62 | 0.85 | 0.87 | 0.88 |
| SS-YOUTUBE | Acc | 0.86 | 0.75 | 0.75 | 0.88 | 0.92 | 0.93 |
| SE0614 | Acc | 0.51 | - | 0.51 | 0.54 | 0.58 | 0.58 |
| SCV1 | F1 | 0.63 | 0.60 | 0.67 | 0.65 | 0.68 | 0.69 |
| SCV2-GEN | F1 | 0.72 | 0.69 | 0.71 | 0.71 | 0.74 | 0.75 |
| ED | Acc | - | 0.43 | 0.40 | 0.46 | 0.46 | 0.48 |
| ED-CUT | Acc | - | 0.41 | 0.36 | 0.42 | 0.44 | 0.45 |

the average was 61, and some workers were definitely contributing more hits than others. To ensure quality, we hand-checked random subsets of conversations by our most-frequent workers. They were allowed to participate in as many of these HITs as they wanted for the first 10k conversations, then we added qualifications to limit the more frequently active workers to a maximum of 100 conversations.

We include ten randomly selected dialogues from our training set in Table 8.

### A.3 EMOTION CLASSIFICATION RESULTS

Our dataset can also be used to train or fine-tune an emotion classifier, as we do in our PREPEND-K and ENSEM-DM+ set-ups. To give a sense of where the difficulty falls compared to existing emotion and sentiment classification benchmarks, we reproduce the table from Felbo et al. (2017) and add results when fine-tuning the Deepmoji model on our dataset, or using a fastText classifier (Table 9).

### A.4 HUMAN RATINGS

### A.4.1 CROWDSOURCING DESCRIPTION

Human evaluations were collected on MTurk. For the rating task, workers were shown one randomly subsampled example from the test set for a randomly selected model (this was done 100 times per model) and asked to rate that single response. 217 US workers participated in the rating task, and had to perform a minimum of one rating. For the human comparison task, workers were shown a dialogue context, and the two responses from a pair of models presented in a randomized order (this was done 50 times per pair of models). They had to select if they preferred one, the other, both equally, or neither. 337 US workers participated in the model comparison task.

### A.4.2 RANKING RESULTS

In Figure 5, we provide the exact comparisons between model responses for the ranking task. Scores less than 1 indicate that the vertical model is preferred, whereas scores greater than one indicate more of a preference for the horizontal model.

| | | Retrieval | | | | | | | | Generation | | | | | | | | Gold |
|---|---|---|---|---|---|---|---|---|---|---|---|---|---|---|---|---|---|---|
| | | pretrain | base | multi | pre1 | pre3 | pre3 | Ens-tran | Ens-dm+ | pretrain | base | multi | pre1 | pre3 | pre5 | Ens-tran | Ens-dm+ | |
| Retrieval | pretrain | 0 | 0.37 | 0.41 | 0.17 | 0.35 | 0.52 | 0.5 | 0.14 | 4.5 | 0.78 | 1 | 1.23 | 0.55 | 0.44 | 0.82 | 0.24 | 0.04 |
| | base | 2.7 | 0 | 1.42 | 0.86 | 1.06 | 1 | 0.94 | 1.07 | 5.17 | 2.33 | 2.3 | 2.56 | 1.19 | 2.62 | 1.5 | 1.62 | 0.18 |
| | multi | 2.44 | 0.71 | 0 | 1.07 | 0.72 | 0.93 | 0.75 | 0.53 | 7.5 | 3.44 | 1.8 | 1.73 | 2.64 | 3 | 1.12 | 1.75 | 0.24 |
| | pre1 | 5.8 | 1.17 | 0.94 | 0 | 2 | 0.88 | 0.95 | 0.94 | 4.17 | 2.25 | 2.15 | 3.71 | 3.86 | 4.5 | 1.54 | 3 | 0.06 |
| | pre3 | 2.86 | 0.94 | 1.38 | 0.5 | 0 | 0.61 | 1.25 | 1.3 | 5 | 1.31 | 3.57 | 3.33 | 1.42 | 1.06 | 1.44 | 1.2 | 0.25 |
| | pre3 | 1.91 | 1 | 1.07 | 1.13 | 1.64 | 0 | 1.07 | 0.82 | 4.83 | 2.4 | 3.29 | 2.33 | 1.47 | 2.3 | 2 | 1.58 | 0.16 |
| | Ens-tran | 2 | 1.07 | 1.33 | 1.06 | 0.8 | 0.93 | 0 | 0.47 | 7.25 | 2 | 2.33 | 2.31 | 1.36 | 1.5 | 1.36 | 1.21 | 0.32 |
| | Ens-dm+ | 7 | 0.93 | 1.88 | 1.06 | 0.77 | 1.21 | 2.11 | 0 | 13.5 | 3.62 | 2.09 | 1.69 | 2.5 | 2.3 | 1.9 | 1.43 | 0.21 |
| Generation | pretrain | 0.22 | 0.19 | 0.13 | 0.24 | 0.2 | 0.21 | 0.14 | 0.07 | 0 | 0.16 | 0.08 | 0.29 | 0.22 | 0.4 | 0.22 | 0.16 | 0.02 |
| | base | 1.29 | 0.43 | 0.29 | 0.44 | 0.76 | 0.42 | 0.5 | 0.28 | 6.25 | 0 | 1.2 | 1.25 | 0.5 | 1.23 | 0.79 | 0.28 | 0.15 |
| | multi | 1 | 0.43 | 0.56 | 0.46 | 0.28 | 0.3 | 0.43 | 0.48 | 12.5 | 0.83 | 0 | 0.78 | 0.79 | 1.06 | 0.57 | 0.36 | 0.1 |
| | pre1 | 0.81 | 0.39 | 0.58 | 0.27 | 0.3 | 0.43 | 0.43 | 0.59 | 3.5 | 0.8 | 1.29 | 0 | 0.47 | 1.4 | 0.3 | 0.4 | 0.02 |
| | pre3 | 1.83 | 0.84 | 0.38 | 0.26 | 0.71 | 0.68 | 0.74 | 0.4 | 4.5 | 2 | 1.27 | 2.12 | 0 | 1.27 | 0.76 | 0.72 | 0.21 |
| | pre5 | 2.25 | 0.38 | 0.33 | 0.22 | 0.95 | 0.43 | 0.67 | 0.43 | 2.5 | 0.81 | 0.95 | 0.71 | 0.79 | 0 | 0.22 | 1 | 0.11 |
| | Ens-tran | 1.22 | 0.67 | 0.89 | 0.65 | 0.7 | 0.5 | 0.74 | 0.53 | 4.6 | 1.27 | 1.75 | 3.29 | 1.31 | 4.6 | 0 | 0.75 | 0.07 |
| | Ens-dm+ | 4.17 | 0.62 | 0.57 | 0.33 | 0.83 | 0.63 | 0.83 | 0.7 | 6.25 | 3.62 | 2.78 | 2.5 | 1.38 | 1 | 1.33 | 0 | 0.11 |
| Gold | | 25 | 5.67 | 4.11 | 16.5 | 4 | 6.4 | 3.09 | 4.67 | 53 | 6.83 | 10 | 45 | 4.86 | 8.75 | 13.5 | 9.25 | 0 |

Figure 5: Full heatmap from ranking task. The prepend models in this tables are from the emotion supervision task. Scores are ratios of the [# times horizontal model is selected over vertical] : [# times vertical model is selected over horizontal]. Scores of greater than 1 indicate a preference for the horizontal model, and scores of less than 1 indicate a preference for the vertical model.

