# OpenReview forum: "I Know the Feeling: Learning to Converse with Empathy"
_ICLR.cc/2019/Conference_

### Official Review · AnonReviewer1 · 2018-10-23
**Attempting to improve chatbot responses with empathy - contributed dataset**

**Rating:** 5
**Confidence:** 3

**Review:**

Overall this paper contributes many interesting insights into the specific application of empathetic dialog into chatbot responses.  The paper in particular is contributing its collected set of 25k empathetic dialogs, short semi-staged conversations around a particular seeded emotion and the results of various ways of incorporating this training set into a generative chatbot.

While the results clearly do not solve the problem of automating emapthy, the paper does give insights into which methods perform better than others (Generation vs Retrieval) and explicitly adding emotion predictions vs using an ensemble of encoders.

There is a lot in this paper, and I think it could have been better organized.
I am more familiar with emotion related research and not language to language translation, so I would have appreciated a better explanation of the rationale for using BLEU scores.  I did some online research to understand these Bilingual Evaluation Understudy Scores and while it seems like they measure sentence similarity, it is unclear how they capture ”relevance” at least according to the brief tutorial that I read (https://machinelearningmastery.com/calculate-bleu-score-for-text-python/).  I did not see the paper describing the use of this score in the references but perhaps I missed it – could you please clarify why this is a good metric for relevance?  It seems that these scores are very sensitive to sentence variation.  I am not sure if you can measure empathy or appropriateness of a response using this metric.
For your data collection you have 810 participants and 24,850 conversations.  Are the 810 participants all speakers or speakers and listeners combined?  How many conversations did each speaker/listener pair perform 32?  (one for each emotion) or 64? (two for each emotion) Was the number variable?  If so what is the distribution of the contribution – e.g. did one worker generate 10,000 while several hundred workers did only three of four?  Was it about even?  Just for clarity – how did you enroll participants?  Was it through AMT?  What were the criteria for the workers?  E.g. Native English speaker, etc.

In your supplemental material, I found the interchanging of the words “context” and “emotion” confusing.  The word context is used frequently throughout your manuscript: “dialog context,” “situational context” - emotions are different from situations, the situational utterance is the first utterance describing the emotion if I read your manuscript correctly.  Table 6 should use “Label” or “Emotion” instead of the more ambiguous “Context.”

My understanding is that speakers were asked to write about a time when they experienced a particular feeling and they were given a choice of three feelings that they could write about.  You then say that workers are forced to select from contexts they had not chosen before to ensure that all of the categories were used.  From this I am assuming that each speaker/listener worker pair had to write about all 32 emotions – is this correct?  Another interpretation of this is that you asked new workers to describe situations involving feelings that had not been chosen by other workers as data collection progressed to ensure that you had a balanced data set.  This would imply that some emotional situations were less preferred and potentially more difficult to write about.  It would be interesting if this data was presented.  It might imply that some emotion labels are not as strong if people were forced to write about them rather than being able to choose to write about them.
Were these dialogs ever actually annotated?  You state in section 2, Related Work “we train models for emotion detection on conversation data that has been explicitly labeled by annotators” – please describe how this was done.  Did independent third party annotators review the dialogs for label correctness?  Was a single rater or a majority vote used to decide the final label.  For example, in Table 1, the label “Afraid” is given to a conversation that could also have reasonable been generated by the label “Anxious” a word explicitly used in the dialog.  I am guessing that the dialogs are just labeled according to the label / provocation word and that they were not annotated beyond that, but please make this clear.
In the last paragraph you state “A few works focus..” and then list 5.  This should rather be “several other works have focused on “ …
Conversely, you later state in section 3 “Speaker and Listener”, “We include a few example conversations from the training data in Table 1,” this should more explicitly be “two.”
Also in section 3 when you describe your cross validation process, you state “We split the conversations into approximately 80/10/10 partitions.  To prevent overlap of <<discussed topics>> we split the data so that all the sets of conversations with the same speaker providing the prompt would be in the same partition.
In your supplemental material you state that workers were paired.  Each worker is asked to write a prompt, which also seems to be the first utterance in the dialog they will start.  You state each worker selects one emotion word from a list of three which is somehow generated (randomly?) form your list of 32 .  I am assuming each worker in the pair does this, then the pair has a two “conversations” one where the first worker is the speaker and another where the second worker is the speaker – is this correct?  It is not entirely clear from the description. Given that you have 810 workers and 24,850 conversations, I am assuming that each worker had more than one conversation.  My question is  - did they generate a new prompt / first utterance for each conversations.  I am assuming yes since you say there are 24,850 prompts/conversations.  For each user are all of the situation/prompts they generate  describing the same emotion context?  E.g. would one worker write ~30 conversations on the same emotion.  This seems unlikely, and it seems more likely that given the number of conversations ~30 per participant is similar to the number of emotion words that you asked each worker to cycle through nearly all of the emotions or that given they were able to select, they might describe the same emotion, e.g. “fear” several times.  If the same worker was allowed to select the same emotion context multiple times was it found that they re-used the same prompt several times?  I am assuming that this is the case and that this is what you mean when you say that you “prevent overlap of discussed topics” between sets when you exclude particular workers.  Is this correct?  Or did you actually look and code the discussed topics to ensure no overlap even across workers (e.g. several people might have expressed fear of heights or fear of the dark).

In section 4, Empathetic dialog generator, you state that the dialog model has access to the situation description given by the speaker (also later called the situational prompt) but not the emotion word prompt.  Calling these both prompts makes the statement about 24,850 prompts/conversations a bit ambiguous.  A better statement would be 24,850 conversations based on unique situational prompts/descriptions (if they are in fact unique situational prompts.  I am assuming they are not if you are worried about overlapping “discussed topics” which I am assuming are the situational prompts since the dialogs are very short and heavily keyed off these initial situational prompts)

In your evaluation of the models with Human ratings you describe two sets of tests.  In one test you say you collect 100 annotations per model.  More explicitly, did you select 100 situational prompts and then ask workers to rate the response of each model?  Was how many responses was each worker shown?  How many workers were used?  Are the highlighted numbers the only significant findings or just the max scores?  Annotations is probably not the correct word here.

Please also describe your process for assigning workers to the second human ratings task.

Since the two novel aspects of your paper are the new dataset and the use of this dataset to create more empathetic chatbot responses ("I know the feeling") I have focused on these aspects of the paper in my review.

I found the inclusion of Table 7 underexplained in the text.  The emotion labels for all these datasets are not directly comparable so I would have liked to have seen more explanation around how these classifications were compared.  It would also be helpful to know how more similar emotions such as "afraid" and "anxious" were scored vs "happy" and "sad" confusions

---

> ### Author Response · Authors · 2018-11-24
> **Response (1)**
>
> Thank you for your thoughtful feedback and pointing out many places where more experimental details or clarifications would be useful, and where we had been inconsistent in our terminology. We addressed your points and incorporated your corrections to the updated manuscript; please find detailed responses below.
>
> “I think it could have been better organized.”: we indeed have extensively re-organized the paper, as detailed in our response in the general thread.
>
> ““ I would have appreciated a better explanation of the rationale for using BLEU scores.  I did some online research to understand these Bilingual Evaluation Understudy Scores and while it seems like they measure sentence similarity, it is unclear how they capture ”relevance” at least according to the brief tutorial that I read (https://machinelearningmastery.com/calculate-bleu-score-for-text-python/).”:: We truly appreciate your thoughtfulness in taking the time to consult background information about BLEU -- we indeed should have included the standard reference for BLEU, Papineni et al [4], which we are adding to the manuscript.
>  We use BLEU as an evaluation metric because it has been frequently used in other dialogue generation papers as an automated evaluation (Li et. al 2016 [1] cited in our manuscript,   Wen et al 2015 [2], Li et al 2015 [3], to name just a few), and we are adding this as well to the manuscript.  However, we definitely agree that word overlap scores do not always align with human judgement, which has been documented in other works such as the “How Not to Evaluate your Dialogue System” paper [Liu et al. 2016] that we mentioned in our discussion of the human evaluation set-up. This is why we include human evaluation in addition to the commonly used automated metrics.
>
> Crowdsourcing process: Yes, we did use Amazon Mturk for recruiting workers, and required  that all of our participants came from the US.  Each pair of workers contributed to at least two conversations, that could have been about the same or different emotion labels, depending on which words they were offered and which words they selected.  Individual workers did not have to have conversations about all 32 emotions; instead, coverage of all emotions was ensured by offering more often the emotions that had been selected less overall. We have added a lot of details about the procedure in the manuscript (in the Appendix A2 and A4 sections, as well as in section 3). As to your specific questions: the median number of conversations per worker was 8, while the average was 61. Thus, there were definitely a handful of workers who were more actively participating. To ensure quality, we hand-checked random subsets of conversations by our most-frequent workers. They were allowed to participate in as many of these HITs as they wanted for the first ~15k conversations, then we added qualifications to limit the more “frequently active” workers to a set number of conversations (100 per worker). We have added that information to the crowdsourcing description in the appendix, as well as a larger random sample of conversations from the dataset.
>
> We would like to clarify what the HITs look like.  In each HIT a worker is first taken to a screen where they are shown 3 emotion words. At first the 3 words were sampled randomly, but as the crowdsourcing data generation process went on, we showed the 3 words that had overall been picked the least so far for a first-time worker, or the 3 that had been used the least for that worker if the worker had already performed the task before, so as to ensure better coverage of all emotion labels. It is true that this makes worker select emotions that they might not spontaneously have preferred, but we observed an initial bias for situations that were easier to describe (e.g, a situation causing surprise), and we thought our dataset would be more useful for training versatile dialogue models if all emotion words were covered in a more balanced way.
>
> “This would imply that some emotional situations were less preferred and potentially more difficult to write about.  It would be interesting if this data was presented. “: While we are not presenting the initial imbalance, or commenting on it in the paper, its residual effect (as well as the fact that workers still had a choice between 3 words, so could effectively exclude ever working on 2) can still be observed in the slight imbalance of our set, in Table 7 of the Appendix, where the labels are ordered by decreasing frequencies.
>
> Workers picked one emotion word among the 3 offered (their own choice) and wrote a description of a time they felt that way. Then, they were taken to another screen where they were paired randomly with another worker who had just completed the same process.  They took turns starting two conversations.  Each worker had to describe their situation as part of the conversation they started.  After that, they answered a few brief feedback questions which helped monitor quality.

---

> ### Author Response · Authors · 2018-11-24
> **Response (2)**
>
> Context vs. emotion:  Thanks for the feedback, the use of “context” for the dialogue utterances that came before the utterance to produce was chosen to be consistent with existing dialogue papers in the literature, so we are keeping it for that specific use and have updated the paper to use  “emotion” or “label” everywhere else.
>
> “which also seems to be the first utterance in the dialog they will start” / “you state that the dialog model has access to the situation description given by the speaker (also later called the situational prompt) but not the emotion word prompt.” / “Calling these both prompts makes the statement about 24,850 prompts/conversations a bit ambiguous. “: thanks for making us realize this was unclear. The situation description does not have to be the first utterance in the dialog started by the Speaker (but it can be if they choose to start the conversation with it, and indeed Speakers chose to do so frequently, about 23% of the time). It is not the case that the model has access to the situation description itself, neither the models nor the Listeners do (unless the Speaker decided to start the conversation with a description matching the situation). We have updated the manuscript in several places to make this important point clear, and also made sure to reserve “prompt” for the emotion word, and “situation description” for the text written by the Speaker in response to this word -- thank you for pointing out that we had used these words in an inconsistent way.
>
> Human ratings/annotations:  indeed our phrasing was confusing, thanks for pointing that out to us. We have updated the manuscript to clarify that the dialogs were generated as prompted by an emotion word and not annotated. As for model scoring, we now use the word “ratings” instead of annotations, as per your suggestion to replace the word “annotation’.
> We added many details about the rating procedure in the Appendix, section A4. To your specific question: we, again, used MTurk.  Workers were shown one randomly subsampled example from the test set for a randomly selected model (this was done 100 times per model) and asked to rate that single response.  217 US workers participated in the rating task, and had to perform a minimum of one rating.
> For the human comparison task, workers were shown a dialogue context, and the two responses from a pair of models presented in a randomized order (this was done 50 times per pair of models). They had to select if they preferred one, the other, or both equally. 337 US workers participated in the model comparison task.
>
> “re-used the same prompt several times”: workers were not prevented from talking about the same situation different times for the same emotion (and indeed some chose to do so, but very rarely), but were paired with different Listeners every time. Our phrasing of “topic overlap” was indeed confusing so we have updated the manuscript to make it clear that we are talking about the same situation being described by the same worker in training and testing set.

---

> ### Author Response · Authors · 2018-11-24
> **Response (3)**
>
> “Are the highlighted numbers the only significant findings or just the max scores?”: in our submission, we had highlighted the maximum score, as is commonly done in papers in this community. But we indeed found that it would make the picture clearer to instead use confidence intervals in the table of human evaluations, so we have now instead highlighted results that were above 2 the standard error of the mean compared to a reference model, as a 95% confidence interval corresponds to 1.96 SEM.
>
> Under-explained addition of Table 7 in the supplementary material / “The emotion labels for all these datasets are not directly comparable so I would have liked to have seen more explanation around how these classifications were compared. “: That table provides context on performance as a benchmark compared to existing sets rather than a new result, to give a sense of relative difficulty compared to existing benchmarks for a given classification system.  We agree that the emotion labels are not directly comparable, but machine learning systems trained on one task can often successfully be fine-tuned to transfer learning between them, e.g. as done in the DeepMoji paper which presents results for all these datasets for a same base architecture.
>
> “It would also be helpful to know how more similar emotions such as "afraid" and "anxious" were scored vs "happy" and "sad" confusions”: Accuracy reported is not weighted by how “bad” the confusion is, so all the confusions are scored the same: classifying an “afraid” situation as “anxious” is penalized as much as classifying it as “happy”. This is a very common trait of supervised benchmarks, although there has been a lot of work on how to improve classifiers and benchmarks by taking into account similarities between labels, e.g. Bengio et al 2010 [5] that construct a data-driven structured hierarchy of labels based on classification performance.
> While it would be very interesting to perform this type of analyses on our data, emotion classification in itself wasn’t the focus on this paper, as emotion classification was mostly used as a way to augment the model with useful representations, so we didn’t include many experiments on emotion classification per se, and leave that for future work.
> As we responded to another reviewer comment, we tried to alleviate potential confusion problems by using more than one label in the prepend setting, and by using an intermediate representations within models that was taken before a determination into a single emotion was outputted, so that there isn’t an information loss caused by the winner-take-all process -- nonetheless, refining classifiers to make better use of label relatedness could be a way to improve representations for our task.
>
> [1] Jiwei Li, Michel Galley, Chris Brockett, Georgios P Spithourakis, Jianfeng Gao, and Bill Dolan. 2016.  A persona-based neural conversation model. In Proc. of ACL. pages 994–1003
> [2] Tsung-Hsien Wen, Milica Gasic, Nikola Mrksic, Pei-Hao Su, David Vandyke, Steve Young. 2015. Semantically Conditioned LSTM-based Natural Language Generation for Spoken Dialogue Systems. In Proc. of EMNLP. pages 1711–1721
> [3]Jiwei Li, Michel Galley, Chris Brockett, Jianfeng Gao, Bill Dolan. 2016. A Diversity-Promoting Objective Function for Neural Conversation Models. In Proc. of NAACL-HLT. pages 110-119
> [4] Kishore Papineni, Salim Roukos, Todd Ward, and Wei-Jing Zhu. 2002. BLEU: a method for automatic evaluation of machine translation. In Proc. of ACL. pages 311–318
> [5] Bengio, S., Weston, J. and Grangier, D., 2010. Label embedding trees for large multi-class tasks. In Advances in Neural Information Processing Systems (pp. 163-171).

---

### Official Review · AnonReviewer3 · 2018-11-03
**A renewed attempt for adapting dialog responses to emotional context**

**Rating:** 7
**Confidence:** 4

**Review:**

The paper describes a new study about how to make dialogs more empathetic.
The work introduced a new dataset of 25k dialogs designed to evaluate the
role that empathy recognition may play in generating better responses
tuned to the feeling of the conversation partner.  Several model
set-ups, and many secondary options of the set-ups are evaluated.

Pros:

A lot of good thoughts were put into the work, and even though the techniques
tried are relatively unsophisticated, the work represents a serious attempt
on the subject and is of good reference value.

The linkage between the use of emotion supervision and better relevancy is interesting.

The dataset by itself is a good contribution to the community conducting studies in this area.

Cons:

The conclusions are somewhat fuzzy as there are too many effects
interacting, and as a result no clear cut recommendations can be made
(perhaps with the exception that ensembling a classifier model trained
for emotion recognition together with the response selector is seen
as having advantages).

There are some detailed questions that are unaddressed or unclear from
the writing.  See the Misc. items below.

Misc.

P.1, 6th line from bottom: "fro" -> "from"

Table 1:  How is the "situation description" supposed to be related to the
opening sentence of the speaker?  In the examples there seems to be substantial
overlap.

Figure 2, distribution of the 32 emotion labels used:
this is a very refined set that could get blurred at the boundaries between similar emotions.
As for the creators of those dialogs,  does everyone interpret the same emotion label the same way?
e.g. angry, furious; confident, prepared; ...; will such potential ambiguities impact the work?
One way to learn more about this is to aggregate related emotions to make a coarser set,
and compare the results.

Also, often an event may trigger multiple emotions, which one the speaker chooses to focus on
may vary from person to person.  How may ignoring the secondary emotions impact the results?
To some extent this is leveraged by the prepending method (with top-K emotion predictions).
What about the other two methods?

P. 6, on using an existing emotion predictor:  does it predict the same set of emotions
that you are using in this work?

---

> ### Author Response · Authors · 2018-11-24
> **Response (1): experiment organization and description**
>
> Thank you for your thoughtful feedback, which was very helpful to improve the paper. Please see our response in the general thread, which details our updates. Regarding your specific notes and questions:
>
> “The conclusions are somewhat fuzzy as there are , and as a result no clear cut recommendations can be made”: thanks for pointing that out -- we have extensively reorganized our paper to make the motivations and results clearer; please also see our response in the general thread.
>
> “How is the "situation description" supposed to be related to the
> opening sentence of the speaker?  In the examples there seems to be substantial
> Overlap.”   This was indeed unclear, thanks for pointing that out -- we have now clarified this. We asked the crowdsourced workers to start the conversation by describing their situation in a conversational way.  Because of this, there often is overlap.  Workers sometimes stuck to the situation description closely, while others were more creative about re-wording things.

---

> ### Author Response · Authors · 2018-11-24
> **Response (2): emotion labels**
>
> “this is a very refined set that could get blurred at the boundaries between similar emotions.”:
>  as mentioned in the paper, we consulted existing works on emotion classification, especially works that had provided previous datasets of similar nature (e.g., Skerry and Saxe 2015). We decided to include all the emotion labels used in those previous works so that people who had used those datasets before could more easily transition to ours, possibly by using only the subset that had those emotion labels. Distinction between similar emotions was not as important to us, since our main focus was generating situations to which Listeners could react with empathy, rather than distinguishing between them. We selected a very fine-grained set of emotion labels so that researchers could group together similar emotions, as needed depending on the application they are interested in (and indeed there is a lot of work on how to cluster labels in a data-driven way, e.g. , Bengio et al 2010 [1]), though we do not try that here. We reasoned that it is easier to group together after the fact than to separate and the focus of this work is not emotion classification.  We also thought that keeping emotions that are similar but suggest some intensity gradation (e.g., angry vs. furious) could even be useful down the line for tasks such as grading emotion intensity, like the task 1 of SemEval 2018.
>
> “does everyone interpret the same emotion label the same way”: no, indeed, but the agreement between humans is high enough to get good signal, and this has been quantified -- for example, see Fig 1C in Skerry and Saxe 2015  (reference in the paper) that finds an accuracy of 65% for 20 labels (all part of our set of 32), where chance would be 5%.
> For an explicit feature-based analysis of similarity, relevant analyses of overlap of features and similarities can also be found in Skerry and Saxe 2015 -- in particular Figure 2 shows how labels (20 labels, that are included in our list of 32) relate to appraisal features (e.g., expectedness, future, familiarity, suddenness, etc), basic emotions, and the affective circumplex.
>
> “ will such potential ambiguities impact the work?
> One way to learn more about this is to aggregate related emotions to make a coarser set,
> and compare the results.” “What about [multitask and ensemble]?”:  With supervised fine-tuning or concatenating representations (the multitask and ensemble settings), the representation used in the model is taken before a single winner is outputted, so there isn’t an information loss caused by the winner-take-all process of outputting a single label  -- however, it is definitely possible that having better clustering of emotions could focus learning on more crucial information than distinguishing whether someone is “angry” or “furious” while they’re actually somewhere in between, and this could be tested in future work, for example in conjunction with existing methods to combine labels. Thanks for the suggestion.
> As you also mention, “To some extent this is leveraged by the prepending method (with top-K emotion predictions).” -- and indeed that was our reason for experimenting with K > 1.
>
> “on using an existing emotion predictor:  does it predict the same set of emotions
> that you are using in this work?”  all of the emotion predictors that we use from other works were trained with different sets of labels than what we use, and not directly emotions (e.g., emojis), however we fine-tune the deepmoji+ model on our set of labels. The deepmoji paper presents many experiments on transferring their emoji classification learning to multiple loosely emotion-related tasks, like sentiment classification (Table 9 in the appendix lists many of those datasets.) One of those datasets is the ISEAR set, which uses labels that we did include in our list and which also starts from short situation descriptions, so we had reason to believe that deepmoji could transfer well to our task.
>
> [1] Bengio, S., Weston, J. and Grangier, D., 2010. Label embedding trees for large multi-class tasks. In Advances in Neural Information Processing Systems (pp. 163-171).

---

> > ### Comment · AnonReviewer1 · 2018-11-26
> > **thanks for your additional comments - I upgraded my rating**
> >
> > thanks for your additional comments - I upgraded my rating.  I am hoping to see even more development based on this dataset and perhaps a longer journal paper in the future.

---

> > > ### Author Response · Authors · 2018-11-26
> > > **Thank you -- your response is on the wrong reviewer thread though**
> > >
> > > Dear AnonReviewer1,
> > >
> > > We are very glad to hear that you found our additional comments useful and upgraded your score to help it get presented to ICLR. We indeed are eager to see a lot more development based on this dataset.
> > > We noticed that you posted your response on the thread of the review of AnonReviewer3, which might be confusing to others -- would you mind reposting on the comment thread of your own review instead?
> > > We would also very much appreciate if you could update your initial review to reflect the new upgraded score.
> > > Thank you again for your time and consideration!

---

### Official Review · AnonReviewer2 · 2018-11-05
**Doubts about the two main contributions**

**Rating:** 4
**Confidence:** 4

**Review:**

The overall goal of the paper is to make end-to-end dialogue systems more empathetic, so that they can respond more appropriately and in ways that acknowledge how the users are feeling. The authors make two contributions towards that goal: (1) they introduce a crowdsourced dataset (EmpatheticDialogue) annotated with fine-grained emotion labels. (2) They show improvements on dialogue generation (in terms of empathy, but also relevance and fluency) using a multi-task objective, ensemble of encoders, and a more ad-hoc technique that consists of prepending inferred emotion labels to the input.

In terms of technical novelty, the work is relatively incremental: (A) The use of multi-task objectives in sequence models [1] is relatively common nowadays (there is little mathematical details in the paper, so it’s hard to see how the approach of the paper really differs from extensive related work.). (B) Prepending predictions: prepending class labels to the input is also relatively common (e.g., in multilingual NMT to select a language). [2] presents a similar approach for polite response generation, where they prepend a label using a politeness classifier.

I also have some doubts about the two claimed contributions of the paper (the authors actually list 3 contributions in the introduction, but for convenience I lump the 2 non-data ones together):

(1) Dataset: The dataset was crowdsourced by giving workers an emotion label (e.g., afraid) and asking them to define a situation in which that emotion might occur and inviting them to have a conversation on that situation. The problem with prompting workers for specific emotions is that this assumes they are good actors and this is likely to produce exchanges that are rather cliché and overdone (e.g., Table 1: the label “afraid” yields a situation that is rather spooky and unlikely in the real world, and the conversations themselves are rather cliché and incorporate little details that would make them sound real).  The authors justify this dataset by pointing out that existing real-world datasets underrepresent rare emotions (e.g., afraid), but that’s just a reflection of how these emotions are distributed in the real world. Better subsampling strategies would enable a better balance in the distribution without having to give up on real-world data (filtering using emojis, hashtags, etc.).  As the paper shows quantitative gains using this dataset, it is probably ok to use but, qualitatively, this dataset is probably not for everyone working on emotion in NLP.

(2) Improvement in empathetic dialogue generation: The paper shows improvements across the board compared to a Transformer baseline, but the question the authors do not satisfactorily address is whether their explicit (and I would say sometimes ad-hoc) treatment of empathy (e.g., using emotion classifier, etc.) is crucially needed to get better empathetic dialogues, since the authors did not control for training data size and model capacity. Indeed, the authors exploited different amounts of data (out of-domain, or both in- and out-of-domain), different model capacities (going from baseline Transformer to model ensembles), and sometimes richer input (e.g., pre-trained emotion classifier). The results might only be showing that more data or more model capacity helps, which would of course not be surprising at all. The fact that generated outputs improve in all aspects (not only empathy, but in attributes completely unrelated to empathy such as fluency and relevance) suggests that the improvement is due to more data or capacity (e.g., perhaps yielding better encoder).  More statistics in the table in terms of number of parameters and amount of in- and out-of-domain data used for each experiment would help draw a clearer picture.

About the use of Reddit: this might not be the best background dataset, as it’s mostly strangers talking to other strangers, presumably causing the baseline to be weak on empathy. Twitter or other social-network type datasets (letting you follow people rather topics) *might* be better suited as it comparatively involves more exchanges between people who actually know each other and who are thus more likely to behave empathetically.

Overall, the paper doesn’t really attempt to make major technical contribution, and instead (1) introduces a dataset and (2) makes empirical contributions, but I think there are problems with both.

Typos:

Introduction: “fro”
References: Elizaa

[1] Minh-Thang Luong, Quoc V. Le, Ilya Sutskever, Oriol Vinyals, Lukasz Kaiser
Multi-task Sequence to Sequence Learning
https://arxiv.org/abs/1511.06114

[2] Tong Niu and Mohit Bansal
Polite Dialogue Generation Without Parallel Data
https://arxiv.org/pdf/1805.03162.pdf

---

> ### Author Response · Authors · 2018-11-24
> **Response (1): dataset**
>
> Thank you for your insightful and detailed review. While we respectfully disagree with some of the points (and will detail why below), we appreciate that in most cases the fault was lying with us not having clarified those arguments in the manuscript, which we have now done. For others, your insightful questions led us to organize our experiments better and supplement them with new ones which collectively make for a clearer picture. To your detailed points, let us start with one of your last observations:
>
> “About the use of Reddit: this might not be the best background dataset, as it’s mostly strangers talking to other strangers, presumably causing the baseline to be weak on empathy. “
> We definitely agree that we would expect Reddit to be weak on empathy, but (1) as stated in our response in the main thread, there isn’t currently an empathy benchmark that we know of to actually quantify that, and (2) we wish for publicly available resources to train a conversation system that would respond with empathy in a reproducible way for the community.
> The Reddit data has the advantage of being easily publicly available, of a very large scale (1.7B comments), and has already been used to train dialogue systems (a few refs in the paper), so it’s good for the publicly available reproducible part. From interacting with Reddit-trained dialogue systems and looking at Reddit data, it indeed doesn’t seem very empathetic. There are two options from there: try and make that system more empathetic, and this is the approach we took in our work. The other one is to look for another background dataset, as you suggest. Unfortunately, there aren’t  many publicly available corpora for training dialogue in a general domain, and their scale is at least an order of magnitude smaller than the Reddit one. We like your specific suggestion of “Twitter or other social-network type datasets (letting you follow people rather topics)”, and we indeed know of existing datasets from Twitter of a large scale, but they have shortcomings. First, tweets have low character limits, which is a constraint that doesn’t match the setting of general dialogue. Second, existing publicly released datasets are orders of magnitude lower in scale, and the conversations are very short. The Twitter corpus from Ritter et al 2010 [1] has 1.3 million conversations, 69% of which have only 2 turns. Sordoni et al 2015 [2] used more than 100 million longer Twitter conversations, but the released Triples Twitter corpus unfortunately has fewer than 5k dialogues.
> A last point that we also clarified in our updated paper (see response in common thread) is that by nature, publicly available datasets from social media are quite different from one-on-one conversations, so even with a better background dataset from public social media, we would want to release data in a one-on-one setting.
>
> “ exchanges that are rather cliché and overdone (e.g., Table 1: the label “afraid” yields a situation that is rather spooky and unlikely in the real world, and the conversations themselves are rather cliché and incorporate little details that would make them sound real).”: the example we had used in Table 1 indeed gives that impression. Thanks for drawing our attention to that; this impression is fortunately not representative of our dataset, which to us seems quite colorful. As stated in the general thread response, we have now added a sample of 10 randomly drawn conversations to give a better sense of what our dataset looks like. That random sample talks about 4 chicken species (Australorps, Rhode Island Reds, Barred Plymouth Rocks, Welsummer), rabies causing sensitivity to light, enchiladas, English majors, too much coffee to start a shift, Tuesday breaks to buy lottery tickets with coworkers, etc.
>
> “existing real-world datasets underrepresent rare emotions (e.g., afraid), but that’s just a reflection of how these emotions are distributed in the real world.”: thank you for making us realize that we had not done a good enough job explaining the limitations of existing datasets for training empathetic conversations. We have updated the manuscript to clarify the nature of existing datasets and why they did not meet our needs. Please also refer to our response in the general thread for more clarifications on this point.
>
> [1] A. Ritter, C. Cherry, and B. Dolan. Unsupervised modeling of twitter conversations. In North American Chapter of the Association for Computational Linguistics (NAACL 2010), 2010.
> [2] A. Sordoni, M. Galley, M. Auli, C. Brockett, Y. Ji, M. Mitchell, J. Nie, J. Gao, and B. Dolan. A neural network approach to context-sensitive generation of conversational responses. In Conference of the North American Chapter of the Association for Computational Linguistics (NAACL-HLT 2015), 2015.

---

> ### Author Response · Authors · 2018-11-24
> **Response (2): experiments**
>
>
> “but the question the authors do not satisfactorily address is whether their explicit (and I would say sometimes ad-hoc) treatment of empathy (e.g., using emotion classifier, etc.) is crucially needed to get better empathetic dialogues [...]”: thank you very much for making that point, and making us realize that our experimental results would benefit from disentanglement. As detailed in the response in the general thread, we have extensively reworked the experimental section to make it clear (1) where benefits in empathy from training on our dataset are seen without any increase in model capacity, (2) why we found it valuable to include experiments combining our base model with external classifiers, (3) how capacity came into play, with new experiments with larger models. Your comments have also led us to downplay the treatment of emotion and add new experiments with topic classifiers, to give a better overall picture.
>
> “More statistics in the table in terms of number of parameters and amount of in- and out-of-domain data used for each experiment would help draw a clearer picture.” We have added a section and table discussing capacity (Table 4) for the crucial experiments. For some of the experiments using a  classifier trained on out-of-domain data, we have clarified our motivation for including them. We do not claim that it is surprising that it should help, rather we aim to provide empirical confirmation of whether it does, for a variety of different data sources, and by how much, so that practitioners can more easily decide which model or data to adopt.
>
> “doesn’t really attempt to make major technical contribution”: we do not argue to the contrary, and it was definitely not how we had tried to cast our contribution. We have updated our manuscript with more citations to previous work, including the ones you provided, to make it even clearer that we claim no innovation on the architecture front -- rather, our goal is to show how existing methods can be used with our dataset, and how they compare.

---

### Author Response · Authors · 2018-11-24
**Paper revision updated -- summary of modifications (1/2)**

We are grateful to all three reviewers for their insightful and thoughtful comments which have helped us substantially improve the manuscript. We uploaded a new version. Here is a summary of the main changes -- other reviewer questions are answered in individual replies to each review.

(1) We clarified why we believe this dataset fills an important need for which no good data exists. AnonReviewer2’s detailed comments on real-world data made us realize that we had not sufficiently explained in the paper the shortcomings of the existing labelled datasets that we are aware of. We changed the Introduction and Related work sections to make that clearer. In particular:
-- we gave more background on DailyDialog, one of the datasets that we discuss as having an extreme skew in emotion labeling. We clarified that DailyDialog was “obtained by crawling educational websites intended for learners of English, includes many dialogues anchored in everyday situations and has been annotated post-hoc with emotion labels, but only ≈ 5% of the utterances have a label other than “none” or “happy”, and dialogues are mostly limited to domains deemed appropriate for use as a language learning tool (ordering from a restaurant, asking for directions, shopping for a specific item, introductions, etc).“  We would respectfully argue that there is no reason to believe that the skews in that dataset actually reflect “real world” distributions: these are the biases of emotions that writers of dialogues for learners of English collectively believe would be the most useful and acceptable as teaching material. Our experience as language learners is that teaching dialogues are often limited to a narrow set of mundane experiences such as asking for directions, introductions, discussing coursework, vacations, etc -- and indeed random sampling of the data yields a lot of examples of these topics.
-- we provided our rationale for preferring crowdsourced data to public social media data, and explained why we believe the biases in public social media data were not a distribution that we should follow for our goal of empathetic conversation: “While public social media content has the advantage of being spontaneous (not elicited) data, it suffers from two shortcomings when used to train a model intended for one-on-one conversation (as opposed to, say, a bot designed to post on Twitter). First, the content is extracted from a context of communication in front of large ”peripheral audiences” (Goffman, 1981) which include potentially everyone with an Internet connection, where the need for curated self-presentation (Goffman, 1959) and the uncertainty as to how wide that audience may be have been shown to lead to different choices of subject matters compared to private messaging, with people sharing more intense and negative emotions through private channels (Bazarova et al., 2015; Litt et al., 2014). Second, Tweets are generally a short-form format limited to 140 characters, which is not a constraint that applies to general conversation.  In this work, we attempt to generate a more balanced coverage of emotions than would appear in public social media content, within a one-on-one framing of unconstrained utterances that is closer to our ultimate goal of training a model for conversation that can respond to any emotion.”
As AnonReviewer2 suggests, it would be better to gather data from conversations between people who know each other -- but because of the nature of public social media communication, we would want those conversations to be from a one-on-one setting.  This makes it impossible to gather and release that type of data from "real interactions" without violating the privacy of users who created it. Crowdsourcing provides dialogues that afford the one-on-one, real-time circumstances, while being much more suitable for reproduction and evaluation of dialogue systems.  Furthermore, by explicitly asking the crowdsourced workers to try and be empathetic, our aim is to create data that captures empathy (as opposed to entertainment value for a public social media audience, etc). We would also like to highlight an observation from R2’s review, that actually contains one of our motivations for doing this work, about how conversation models trained on Reddit would “presumably [...] be weak on empathy.” We would indeed expect that, but to the best of our knowledge there currently aren’t empathy benchmarks, and we found very little previous work on measuring empathy. Our work tries to remedy that.
-- comments from reviewers made us realize that including only two dialogues did not give a good sense of the dataset. We included 10 additional dialogues picked completely randomly from our data; we only rejected samples that created formatting problems in the text. The larger sample is presented in Table  8 of the appendix. We hope that the colorful sample makes it clear that our dataset is not the set of cliches with no details that AnonReviewer2 feared.

---

### Author Response · Authors · 2018-11-24
**Paper revision updated -- summary of modifications (2)**

(2)  We organized our experiments better, as comments from all three reviewers helped us see that a clearer organization would greatly benefit the paper. We have now clearly separated the experiments in two sets:
-- experiments showing that using our data to train models improves the performance of a conversation model trained on Reddit on multiple dimensions, without using any other type of data or additional models, or increasing the capacity of the model beyond 0.01%, with the goal of demonstrating how our data can help create better conversation models
--  experiments comparing many ways to combine a pretrained conversation model and external pretrained models so as to bank previously conducted training without having to conduct costly retraining. We hope to help practitioners who would want to combine our dataset with existing models get a sense of what empirically works better or not on this benchmark.
AnonReviewer2 made an insightful point regarding the need to be more precise about model capacities, and clearer as to where additional data / capacity was used. In addition to the new clear partition of experiments, we have added a table with resource and parameter counts (table 4), and a paragraph to discuss capacity in section 4.1, with new experiments with larger models.
We also added experiments using supervision from topic classification instead of emotion supervision (results added to Tables 2 and 3), and downplayed the focus on emotion to instead emphasize that many other types of good representations could be leveraged.
(3) Comments from all three reviewers helped us considerably clarify the experimental procedure throughout the paper and with a much longer section in the appendix, to use terminology more consistently, and present the main experimental results in a clearer way.

We hope that reviewers will find that we have adequately addressed their thoughtful comments, and that these extensive improvements to our manuscript will convince them to raise their scores so that we can share our work and dataset with the community.

---

### Meta-Review · Area_Chair1 · 2018-12-20

**Confidence:** 5
**Recommendation:** Reject

**Metareview:**

The reviewers raised a number of concerns including the usefulness of the presented dataset given that the collected data is acted rather than naturalistic (and the large body of research in affective computing explains that models trained on acted data cannot generalise to naturalistic data), no methodological novelty in the presented work, and relatively uninteresting application with very limited real-world application (it remains unclear whether having better empathetic dialogues would be truly crucial for any real-life application and, in addition, all work is based on acted rather than real-world data). The authors’ rebuttal addressed some of the reviewers’ concerns but not fully (especially when it comes to usefulness of the data). Overall, I believe that the effort to collect the presented database is noble and may be useful to the community to a small extent. However, given the unrealism of the data and, in turn, very limited (if any) generalisability of the presented to real-world scenarios, and lack of methodological contribution, I cannot recommend this paper for presentation at ICLR.